# ICP-Mass-Spectrometry Ionic Profile of Whole Saliva in Patients with Untreated and Treated Periodontitis

**DOI:** 10.3390/biomedicines8090354

**Published:** 2020-09-15

**Authors:** Federica Romano, Alexandra Castiblanco, Francesca Spadotto, Federica Di Scipio, Mery Malandrino, Giovanni Nicolao Berta, Mario Aimetti

**Affiliations:** 1Department of Surgical Sciences, Centro Interdipartimentale di Ricerca (C.I.R.), Dental School, Section of Periodontology, University of Turin, 10126 Turin, Italy; federica.romano@unito.it (F.R.); alexaca91@hotmail.com (A.C.); spadotto.francesca@gmail.com (F.S.); 2Department of Clinical and Biological Sciences, University of Turin, 10043 Orbassano (To) Turin, Italy; federica.discipio@unito.it; 3Department of Chemistry and Centre for Nanostructured Interfaces and Surfaces (NIS), University of Turin, 10125 Turin, Italy; mery.malandrino@unito.it

**Keywords:** metals, periodontitis, saliva, inductively coupled plasma mass-spectrometry

## Abstract

Over the past decade, there has been growing interest in the association between macro and trace minerals in body fluids and systemic diseases related to chronic inflammation and oxidative stress. Due to the paucity of data in the literature on periodontitis, the aim of this cross-sectional study was to assess the relationship between mineral elements in saliva and periodontal status in patients with untreated and treated periodontitis compared to periodontally healthy controls. Salivary samples from 66 nonsmoker healthy patients (20 periodontally healthy, 24 untreated severe periodontitis and 22 treated severe periodontitis) were analyzed by using inductively coupled plasma mass-spectrometry (ICP-MS). Significant increases in copper (Cu), sodium (Na), iron (Fe) and manganese (Mn) concentrations occurred in saliva of severe periodontitis subjects compared to periodontally healthy controls. No differences were detected between healthy controls and treated periodontitis patients apart from levels of zinc (Zn) and lithium (Li) that were found to be increased and reduced, respectively, in periodontitis group. Most subjects were correctly separated by cluster analysis into active periodontitis and periodontally healthy individuals. Treated periodontitis individuals were classified as healthy subjects. Based on these preliminary results, the assessment of salivary concentration of mineral elements might be useful in discriminating periodontal health and disease.

## 1. Introduction

Periodontitis is a widespread chronic inflammatory disease affecting tooth-supporting tissues, initiated and propagated through a complex interaction between bacterial biofilm and host defense system [1]. The shift from a symbiotic microflora to a dysbiotic pathogenic community triggers the host inflammatory response that contributes to the periodontal breakdown [2]. As periodontitis progresses, gingival inflammation, soft tissue destruction and alveolar bone loss occur sequentially. Meanwhile, inflammatory mediators, tissue destructive molecules, and metabolites are released into the oral fluids [3]. Thus, qualitative and quantitative changes in the composition of oral fluids could have diagnostic and therapeutic significance [4].

Saliva has the capability to mirror both oral and systemic health conditions since it contains a highly complex mixture of substances originating from multiple local and systemic sources [5]. Its collection is noninvasive, low cost and simple to perform and does not cause patient discomfort [4]. Several biomarkers, mainly locally produced proteins of host and bacterial origin (enzymes, immunoglobulins and cytokines) and markers of oxidative stress, have been already identified as correlating with the clinical parameters of periodontitis [6,7,8], but none of them, taken individually, can give a complete scenario of the clinical situation.

Previous investigations documented a close correlation between serum levels of metal ions and systemic diseases related to chronic inflammation and oxidative stress like cardiovascular diseases, obesity, metabolic syndrome and type 2 diabetes [9,10,11,12]. In spite of the increasing evidence of the role of metal ions in periodontal health, little information is available on the changes in salivary ionic profile during periodontal diseases [13,14,15,16]. Saliva contains a variety of macro and trace minerals that are involved in periodontal hard and soft tissue homeostasis and regulate levels of inflammatory cytokines and oxidative stress markers that are responsible of periodontal tissue damage and impaired resistance to infection [17,18]. In particular, it has been previously reported that trace minerals like zinc (Zn), copper (Cu) and manganese (Mn) can be involved in the immune-inflammatory pathways as constituents of antioxidant enzymes [19,20], and major elements like sodium (Na) and potassium (K) regulate the structural integrity of epithelial and connective cells [21]. In addition, magnesium (Mg), calcium (Ca), iron (Fe) and rubidium (Rb) have been identified as elements involved in the modulation of alveolar bone remodeling and periodontal ligament homeostasis [17,22,23], and lithium (Li) as stimulus for bone regeneration [24].

In the light of these findings, it could be hypothesized that salivary levels of metal ions may be indicative for periodontal tissue health and disease. In addition, the examination of the ionic profile in periodontitis patients before and after the completion of active periodontal treatment may provide new insights in periodontitis pathogenesis and may have diagnostic value. Among methodologies applied to elemental profiling in human saliva, to perform these objectives, we employed inductively coupled plasma mass spectrometry (ICP-MS). This is highly a sensitive technique that allows multielement determination in minimal biological sample volumes [25].

Thus, the aim of this cross-sectional study was to assess the relationship between levels of mineral elements in saliva and periodontal status in patients with untreated and treated periodontitis compared to healthy controls by ICP-MS technique.

## 2. Experimental Section

### 2.1. Study Group

A consecutive sample of 66 individuals, aged between 30 and 65 years, was selected among patients seeking oral health consultation at C.I.R Dental School, University of Turin (Italy) in the period between May 2019 and December 2019. This study was approved by the Institutional Ethical Committee of the AOU Città della Salute e della Scienza di Torino (approval No 0050509 on 16 May 2019) and was conducted according to the declaration of Helsinki. All enrolled patients expressed informed consent to take part in the study and signed the appropriate form.

Patients were divided into three groups according to their periodontal status. Twenty-four patients (17 men and 7 women, mean age of 55.4 ± 13.8 years) suffered from untreated generalized periodontitis stage III or IV grade B or C according to the clinical and radiographic criteria of the new classification of periodontal diseases [26]. These patients had not undergone periodontal treatments in the year prior of the enrolment (untreated periodontitis). A second group of treated periodontitis patients comprising 22 subjects (13 men and 9 women, mean age of 53.1 ± 14.6 years) with a past diagnosis of severe generalized periodontitis who had completed the recommended comprehensive periodontal treatment plan and presented a reduced but stable periodontium (no sites with probing depth (PD) ≥ 4 mm with bleeding on probing, and full mouth bleeding score (FMBS) < 10%) [27].

A third group consisting of 20 periodontally healthy subjects (control group, 12 men and eight women, mean age of 44.3 ± 10.8 years) who showed intact periodontium without evidence of periodontal disease, all PDs were ≤3 mm, FMBS < 10% and no radiographic evidence of bone loss.

Subjects with BMI ≥ 30 kg/m^2^, suffering from systemic disease (coronary heart disease, diabetes mellitus, arthritis and malignant disease, HIV infection, renal disease, hepatic disease, immunological diseases, organ transplantation), being treated with drugs that cause an increase in the gingival volume (such as calcium channel blockers, immunosuppressants, antiepileptics, estroprogestinics) or with antibiotic or anti-inflammatory therapy for the last three months, smokers, alcohol consumers, pregnant or breastfeeding women were excluded from this study. Moreover, individuals occupationally exposed to metals or using dietary supplements were also excluded.

### 2.2. Periodontal Examination and Saliva Collection

Each subject underwent a full-mouth periodontal probing, along with intraoral periapical radiographs using a long-cone technique. The following periodontal parameters were recorded by two calibrated examiners with a manual periodontal probe with 1-mm markings (UNC-15, Hu-Friedy^®^, Chicago, IL, USA) on six sites per tooth: presence/absence of bacterial plaque, presence/absence of bleeding on probing, PD as distance from the gingival margin to the location of the tip of the periodontal probe inserted in the sulcus/pocket, and clinical attachment level (CAL) as the distance from the cementoenamel junction to the location of the inserted tip. Gingival inflammation was expressed as percentage of bleeding sites of the total number of sites in the dentition (FMBS). Plaque index was calculated as percentage of sites harboring plaque (FMPS).

After periodontal examination, the subjects were rescheduled for saliva collection. Unstimulated 5 mL whole saliva sample was collected from each subject between 8 am and 10 am to standardize the collection according to the circadian rhythm [28]. Patients were instructed not to drink alcohol in the previous 12 h, and to avoid food, sugar drinks and caffeine in the morning of the collection of the samples. They were also asked not to use toothpastes and mouthwashes during home oral hygiene in order to avoid alterations in the compositions of saliva.

The subjects were then asked to let the saliva pool in their floor of the mouth to their maximum extent while seated in an upright position and then expectorate into a graduated sterile tube for about 5–10 min. Collected samples were immediately frozen at −80 °C until analyzing.

### 2.3. Laboratory Analysis

The saliva samples were centrifuged at 3500 rpm for 10 min. An aliquot of 500 µL was subsampled from the supernatant of each sample and, then, diluted at 5 mL using a 1% *v/v* sub-boiled nitric acid solution [29,30]. A Thermo Finnigan Element 2 sector field inductively coupled plasma mass spectrometry (SF-ICP-MS) system, equipped with a glass concentric nebulizer and a Twinnabar (cyclonic) spray chamber, was used for the measurements of the following elements: Li, Na, Mg, Mn, Fe, Cu, Zn, Ca, K, and Rb.

Mass resolution and isotope selection were optimized for each element to ensure resolution of spectral interferences and maximize sensitivity. Analyses on each sample were conducted following a 60 s uptake and stabilization period. Between samples the nebulizer system was rinsed for 1 min with 2% sub-boiled HNO_3_, which eliminated carry-over and reconditioned the sampler cone. Power applied was 1270 W, 1 L/min flow of both auxiliary and nebulizer gasses, while plasma gas was fluxed at 16 L/min. The limit of detection (LOD) for each element was defined as three times the standard deviation of the blank samples [31].

### 2.4. Statistical Analysis

Statistical analyses were performed by the SPSS (SPSS, version 24, Chicago, IL, USA) statistical program. The Shapiro–Wilk test was used to verify the Gaussian distribution of quantitative variables. Differences in the concentration of the various mineral elements in saliva among the three experimental groups were evaluated using the Kruskal–Wallis test. In case of statistical significance, multiple comparisons were performed by post hoc Dunn test. Differences among the groups in terms of gender, age, FMPS, FMBS, PD and number of teeth were analyzed by a chi-square test and one-way analysis of variance (ANOVA), followed by post hoc Tukey test.

The statistical significance of the relationship among metal ions was determined using the Spearman rank correlation coefficient and the principal component analysis (PCA), with Varimax rotation, which condenses variables that are highly correlated into a set of factors. Data were normalized before analysis. Finally, an exploratory hierarchical cluster analysis (HCA) using the Ward’s hierarchical method of agglomeration was performed to classify salivary samples based on their chemical element concentration. According to Formann, who recommended the number of variable (m) of 2^m^ = sample size, five variables were inserted into the model [32]. Clustering was based on the squared Euclidean distance as a measure of the distance between the observations. The results were reported in the form of dendrogram. Statistical significance was set at 5%.

## 3. Results

There were no statistically significant differences between the three experimental groups in gender distribution (*p* = 0.654). With regard to age, there were no statistically significant differences when comparing healthy controls with untreated periodontitis patients (*p* = 0.058) and both periodontitis groups (*p* = 0.824), while a statistically significant difference emerged between healthy controls and untreated periodontal patients (*p* = 0.007).

Clinical parameters are presented in Table 1. As expected, mean values of all periodontal parameters were significantly higher in untreated periodontitis group than in control group (*p* < 0.001). Patients who successfully completed the periodontal active therapy presented clinical conditions comparable to periodontally healthy subjects in terms of average FMPS, FMBS and PD values but tended to have fewer teeth.

### 3.1. ICP–MS-Based Ionic Profiling in Saliva

Table 2 summarizes the distribution of the chemical elements analyzed using the ICP-MP technique in saliva. Irrespective of the periodontal conditions, Na revealed the highest mean levels in saliva followed by Ca and Mg.

No statistically significant differences were observed between the values detected in the three experimental groups for Mg (*p* = 0.356), K (*p* = 0.858), Ca (*p* = 0.633), and Rb (*p* = 0.899). The salivary levels of Na, Mn and Fe were found to be statistically significantly different between groups (*p* = 0.002, *p* = 0.016 and *p* = 0.008, respectively). Patients with untreated periodontitis had values above those of healthy controls and patients with restored healthy periodontium at the completion of active periodontal therapy, while levels of treated periodontitis patients were not statistically different from those of healthy controls. The salivary levels of Zn significantly reduced (*p* = 0.010) after treatment, while the levels of Li increased (*p* = 0.037). Although Cu was higher in saliva of patients with active disease than in treated periodontitis and healthy patients, the difference did not reach statistical significance due to the large variability between subjects.

### 3.2. Correlation Analysis

Table 3, Table 4 and Table 5 summarize the metal-to-metal correlations in the oral fluid of periodontitis patients before and after periodontal treatment and healthy controls.

For patients with untreated periodontitis (Table 3) Cu was positively correlated with many metals: Na (r = 0.483), Mg (r = 0.673), Ca (r = 0.440) and Mn (r = 0.527). Other relevant correlations were Zn-K (r = 0.475), Mg-Na (r = 0.457) and Mg-Ca (r = 0.453).

For periodontitis patients who had completed the active phase of the therapy (Table 4), most of the mutual relationships observed before the periodontal treatment were still present. Positive correlations were also observed between Fe and four chemical elements: Na (r = 0.495), Mg (r = 0.734), Cu (r = 0.493) and Zn (r = 0.463). In addition, K correlated with Na (r = 0.557), Mg (r = 0.609) and Mn (r = 0.456).

In the control group (Table 5) strong positive correlations were again found between Cu and three metals: Na (r = 0.573), Mg (r = 0.558) and K (r = 0.550). Additionally, some significant correlations were also observed between Mg and two elements: K (r = 0.669) and Zn (r = 0.526).

The interactions between the salivary concentrations of chemical elements was also analyzed by means of the PCA. Figure 1 shows the score plots for the first two PCs (PC1–PC2). In the untreated periodontitis group (Figure 1A), positive associations were observed between Ca-Rb, Mg-Rb, Ca-K and Zn-Mg. In the treated periodontitis group (Figure 1B), positive correlations were found between Zn-Mn, Ca-Na, K-Rb, while in the healthy control group between Li-Mg, Na-Cu, and Zn-Rb. Negative association was detected between Ca and Mn. A visual examination of Figure 1 showed that the pattern of mutual dependence of metals in saliva of untreated periodontitis patients was different from those of both treated periodontitis patients and periodontally healthy individuals.

### 3.3. Cluster Analysis

An exploratory HCA was performed to examine whether it was possible to separate periodontitis patients before and after treatment from periodontally healthy subjects on the basis of their mineral content of saliva. The variables considered for clustering were the salivary concentrations of the elements that demonstrated a statistically different distribution among the experimental groups: Li, Zn, Na, Mn, and Fe. The corresponding dendograms are presented in Figure 2. It can be seen (Figure 2A) patients with untreated severe periodontitis (with three exceptions) and subjects with periodontally healthy conditions (with four exceptions) included in two different clusters. Severe periodontitis patients who had stable but reduced periodontium at the completion of active periodontal treatment were not separated from healthy controls (Figure 2B).

## 4. Discussion

Periodontitis is the result of a chronic immune-inflammatory process to the microbial challenge that leads to oxidative destruction of tooth-supporting tissues and release in the biological fluids of inflammatory markers and breakdown products [1,2,3]. Within this context, changes in ionic content of whole saliva would be indicative of the actual presence and severity of periodontitis as well as of treatment response. Mineral ions are biologically important for the maintenance of periodontal tissues in healthy state due to their ability in regulating immunity and inflammatory mechanisms and in modulating hard and soft oral tissue homeostasis [17].

To the best of our knowledge, this study is the first to consider the salivary concentration of a large number of mineral elements and, in particular, in patients with untreated and successfully treated severe periodontitis in comparison with healthy controls. Of the mineral elements targeted in the current study, Na, Mn, Fe and Cu were significantly increased in untreated severe periodontitis patients compared to both treated periodontitis and healthy groups. Periodontal treatment reestablished a mineral composition of saliva similar to that of periodontally healthy subjects apart from Li and Zn that increased and reduced, respectively, compared to controls.

In agreement with the present results, Na levels were found higher in the saliva of patients suffering from chronic or aggressive periodontitis compared to patients with gingivitis [33] and healthy periodontium [33,34,35]. However, while some studies found a correlation between severity of periodontal damage in terms of PD or CAL and salivary levels of Na [35,36] others failed to confirm it [34]. It is known that Na is one of the most abundant minerals in the body, mainly distributed in blood, bone and connective tissues. It regulates the passage of fluids and nutrients out of cells and participates in the transmission of the impulses of the nerves. It is likely that the increase in Na in periodontitis patients could be a consequence of bone resorption. In spite of the large quantity of Na in bone tissue, only a small part enters into immediate exchange with that in the extracellular spaces. Following the destruction of the alveolar bone tissue, it can be released into the extracellular compartment and into the gingival crevicular fluid (GCF) [21]. It has been also reported that Na concentration tends to increase proportionally according to the severity of the inflammatory status [37]. The volume of GCF increases as inflammation becomes more severe and periodontal damage worsens [38], thus higher salivary Na concentration may reflect the destructive inflammatory reaction within periodontal environment.

The increase in Mn concentrations in active periodontitis is in line with data from previous studies [15,33], but is in contrast with the findings by Huang et al., who observed significant decrease in Mn levels in saliva of periodontitis patients compared to healthy controls. According to the authors, the significant increase in Mn (but also in Cu and Zn) is consistent with a decreased level of the antioxidant superoxide dismutase (SOD) in saliva [39]. Manganese is essential for bone health homeostasis, including bone development and maintenance [17]. It is also incorporated into SOD, which protects against free radicals involved in periodontal tissue damage [18,19]. It is possible that increased levels of Mn indicate that antioxidant protective enzymes are not working well leading to progression of periodontitis. Conversely, it can be hypothesized that the formation of reactive oxygen radicals (ROS) has stimulated a release in the oral fluids of Mn to fulfill its antioxidant function.

Analyzing the trend of Fe, it was possible to find a significant increase of this element in severe periodontitis when compared to periodontal health conditions considering both intact and reduced periodontium. While this finding is supported by the study of Inonu et al. [33], it was in contrast with data from Herman et al. and Huang et al. in saliva [15,39]. Increased Fe serum levels have been observed in diabetic patients with periodontitis [12,40]. Iron regulates erythropoiesis and the formation of hemoglobin, maintaining a correct concentration of this element is a prerequisite for periodontal health [41]. Increased levels of free Fe may act as pro-oxidant and catalyze Fenton reactions leading to the formation of ROS from neutrophils and macrophages [17]. This in turn causes the activation of matrix metalloproteinases and release of proinflammatory cytokines involved in periodontal tissue destruction and alveolar bone resorption [17]. Elevated amounts of Fe in the extracellular environment play also an essential role in regulating the growth and virulence of some anaerobic periodontopathogenic bacteria such as *Porphyromonas gingivalis*, *Treponema denticola* and *Prevotella intermedia* [42,43]. Finally, according to Groenink et al. increased Fe concentration in GCF may reflect elevated levels of hemoglobin leaking from the blood vessels within the inflamed connective tissue in the periodontal pocket wall [44]. The concentration of hemoglobin in saliva or oral rinse using water has demonstrated positive association with degree of gingival inflammation, periodontal parameters, and bone loss [45]. For these reasons the salivary occult test has been proposed for periodontal diagnosis, but has not gained widespread use due to the false positive results in healthy subjects and the inability to selectively detect periodontitis patients [46].

The present results for the salivary concentration of Cu were in agreement with those reported by other authors [13,14,15,16], who also observed higher levels of this metal in periodontitis than in periodontally healthy conditions. However, due to the large variation among periodontitis patients the difference with healthy controls did not reach statistical significance. Copper is a constituent of different enzymes and proteins that regulate lipid and Fe metabolism and connective tissue synthesis. It is important for the correct functioning of the immune system and for the modulation of the antioxidant status. Since it is involved in the conversion of superoxide to hydrogen peroxide and hydroxyl radicals, increased Cu levels can lead to increased oxidative stress, which can cause periodontal tissue destruction [47]. Elevated Cu serum levels alter collagen metabolism and host defense mechanisms against infections by impairing both neutrophil functions and proliferation and antigen-specific antibody production, thus increasing the susceptibility to periodontitis [48,49].

Interestingly, serum levels of ceruloplasmin, an acute phase reactant with ferroxidase activity and involved in Cu transport, appear significantly elevated in periodontitis when compared with those from healthy controls [50]. This finding may suggest a defensive response to the oxidative stress induced by periodontal inflammation.

As suggested by Boras et al. [16], it is possible that increased salivary levels are also associated to greater antimicrobial activity against periodontopathogens. Cu inhibits the colonization of the subgingival environment by *Porphyromonas gingivalis* favoring the adhesion of salivary and serum proteins on its cell surface and thus inhibiting its coaggregation [51].

There is a lack of data in the literature on the mineral profiling of saliva in periodontitis patients who, at the completion of active periodontal treatment, had restored clinical gingival health on a reduced periodontium but they are still susceptible to periodontal disease. On this group of patients clinical parameters of inflammation are under control and the compliance is high, as it is evident in the percentage of FMPS and FMBS < 10% and comparable to the values found in the control group in periodontal health condition.

It is noteworthy that gingival health on an intact or a reduced periodontium had similar ionic salivary composition apart from Zn and Li. Evidence on the role of Zn in periodontitis is contradictory. While some studies reported lower concentration in the saliva of periodontitis subjects [14,39] other observed similar [15,52] or higher amount compared to controls [33]. Zinc is an element that is found inside the oral cavity (dental plaque, hard dental tissues and saliva) and has a fundamental role in maintaining oral health. It exerts an antioxidant action controlling the formation of ROS, regulates the function of immune cells and the secretion of cytokines and plays a central role in collagen tissue formation and bone metabolism [53]. It can be hypothesized that the decreased levels observed after active periodontal treatment could be attributed to a redistribution of the metal after the resolution of the inflammatory process and a higher uptake by the gingival cells. At the molecular level, Zn enhances the expression of Zn-finger protein A20, which inhibits the synthesis of tumor necrosis factor alpha and interleukin-1 beta, the main cytokines associated with periodontal tissue damage [54]. Furthermore, periodontal treatment reduces the salivary levels of zinc-dependent matrix metalloproteinases (MMPs), tissue-destructive enzymes that are responsible for the extracellular matrix degradation in periodontitis. Controlling these inflammatory mechanisms contributes to inhibiting the progressive tissue breakdown and collagen depletion in the periodontal tissues [55].

Recently, it has been suggested the Cu:Zn ratio to be a better indicator of inflammatory status than Cu and Zn alone [17]. Consistently with previous studies on saliva and serum [13,40], periodontitis patients exhibited higher Cu:Zn ratio that reduced at values similar to those of healthy controls after treatment.

Analyzing the trend of Li, it was possible to find a significant increase of this element in patients successfully treated for periodontitis compared to untreated periodontitis and healthy controls. The role of Li in periodontal disease has not yet been addressed. Previous investigations have demonstrated that it promotes cell proliferation and cementogenic/osteogenic differentiation of periodontal ligament cells which are crucial in periodontal tissue regeneration [56,57]. Thus, higher Li levels may modulate periodontal repairing processes.

Given the complex interactions among ions in saliva, the analysis of the pathway of a single metal ion may lose information. In this context, the correlation analysis confirmed the different mutual dependences of the targeted ions in the saliva of the three periodontal groups. We performed a cluster analysis to classify periodontal health and disease based on the salivary ionic content. Herman et al. successfully applied such a statistical approach in distinguishing chronic periodontitis from periodontal healthy status [15]. In the current study, cluster analysis enabled the correct division of healthy controls from untreated periodontitis patients but not from those with successfully treated disease. Subjects with clinical gingival health restored after active treatment were grouped together with healthy controls.

Here, we are reporting for the first time, that periodontitis subjects display a distinctive salivary ionic profiling compared to both healthy and periodontally treated subjects. This may have relevant implications into the clinical practice. Nowadays, periodontal diagnosis is based on radiographic and clinical measurements that are time- and money-consuming and are indicators of previous periodontal tissue breakdown rather than present disease activity. Moreover, they are inadequate to measure the degree of susceptibility to disease progression [8]. Due to its chronic nature, periodontitis progresses without causing severe discomfort for the patient who, often, realizes that he is affected when the disease is at an advanced stage and requires complex and expensive treatment. Thus, it is compelling to explore new, safe, quick, and easily accessible approaches to discriminate current periodontitis from past periodontitis and healthy conditions [58]. For these reasons, many analytical techniques have been used in the last years to discover early markers of periodontitis in saliva [59].

Our work follows this trend: it has produced promising results. Analysis of saliva samples will aid constructing a reliable tool to monitor current disease activity and the response to treatments, in the view of a personalized and precision medicine. Moreover, our study not only proposes ions as biomarkers of periodontal disease, but it could offer a new innovative approach to treat periodontal disease with molecules that chelate or release ions. Obviously, it is necessary to know more about the pathogenesis of periodontitis and the role that every single mineral plays. Hence, it is possible to consider the ionomics not only as a biomarker of periodontitis but also as a possible therapeutic target: a set of chelators and ionophores able to remove, redistribute and deposit ions.

We are aware of the limitations in our approach. First of all, the cross-sectional nature of this study that prevents us from establishing causal relation. However, the similarity of ionic profile in treated periodontitis patients and healthy controls would suggest that elevated levels of metal ions in saliva are associated to the inflammatory status rather than be a risk factor for periodontitis. Certainly, prospective studies would add more valuable information. It should be also considered that we could not discriminate between host cells and bacteria related products. Saliva accumulates elements of the metabolism of both bacterial and host origin from the GCF [38].

Moreover, the small sample size may have prevented some differences to reach statistical significance. A larger sample size would also enable the development of a clinical algorithm. We intend to extend the research to achieve this goal within the next future. Finally, the salivary content in macro and trace elements may vary with the dietary element intake.

## 5. Conclusions

Based on our results, the assessment of salivary concentration of mineral elements might be useful in discriminating periodontal health and disease. Na, Mn, Fe and Cu follow a distinctive trend in conditions of health and periodontal pathology suggesting the existence of ionic-related perturbations inside the oral cavity of patients suffering from periodontitis. The facts that the element profile in saliva of periodontitis patients changes at the end of the therapeutic phase allows us to understand how a clinically effective treatment is able to re-establish a state of homeostasis. Indeed, it is important to underlie that such biological matrix is considered to reliably reflect both oral and systemic health conditions. Because collecting saliva involves noninvasive method and due to the fact that it is an abundant and easily accessible biofluid, element analysis of whole saliva may be a promising tool for diagnostic purposes and treatment monitoring.

## Figures and Tables

**Figure 1 biomedicines-08-00354-f001:**
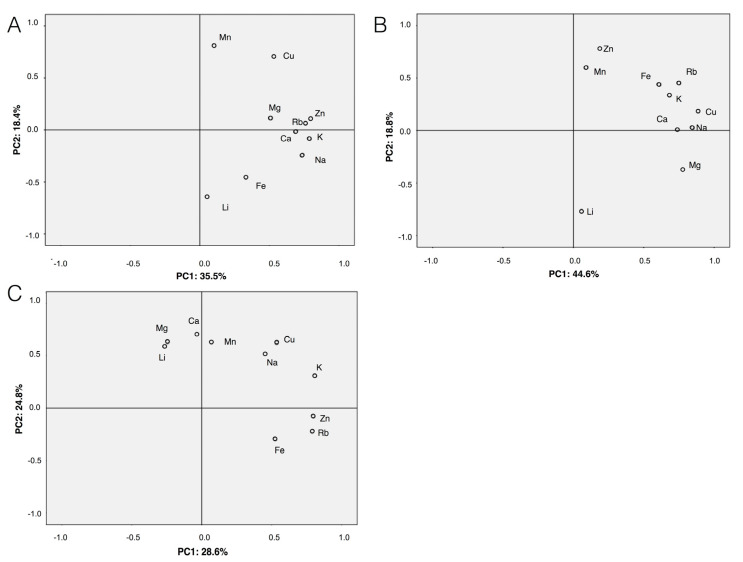
Projection of concentrations of chemical elements in saliva of patients with untreated periodontitis (**A**), with treated periodontitis (**B**) and healthy controls (**C**) on the plane of the first two principal components (PC1 and PC2).

**Figure 2 biomedicines-08-00354-f002:**
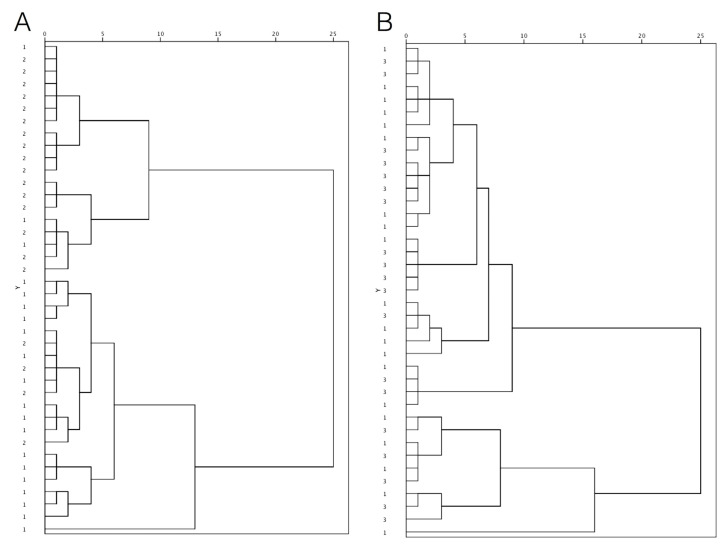
The dendogram obtained from saliva samples: (**A**) clustering of patients with untreated periodontitis (group 2) and healthy controls (group 1); (**B**) clustering of patients with treated periodontitis (group 3) and healthy controls (group 1).

**Table 1 biomedicines-08-00354-t001:** Clinical characteristics of the experimental groups (mean ± SD).

Variables	Healthy Controls (*n* = 20)	Untreated Periodontitis (*n* = 24)	Treated Periodontitis (*n* = 22)
Number of teeth	28.3 ± 1.8	25.5 ± 4.4	26.1 ± 2.7
FMPS (%)	12.8 ± 3.6	73.0 ± 18.4	13.6 ± 4.1
FMBS (%)	7.9 ± 2.0	69.1 ± 24.6	8.4 ± 1.6
PD (mm)	2.4 ± 0.3	3.8 ± 0.7	2.7 ± 0.4

FMPS: full-mouth plaque score; FMBS: full-mouth bleeding score; PD: probing depth.

**Table 2 biomedicines-08-00354-t002:** Concentration of mineral elements in saliva of periodontitis patients and healthy controls.

Mineral Elements	Healthy Controls (*n* = 20)		Untreated Periodontitis (*n* = 24)		Treated Periodontitis (*n* = 22)	
	Mean ± SE	Median (range)	Mean ± SE	Median (range)	Mean ± SE	Median (range)
Li (μ/L)	0.99 ± 0.17	0.87 (0.02–2.80)	1.19 ± 0.21	1.10 (0.15–4.13)	1.78 ± 0.27 ^A^	1.47 (0.62–6.84)
Na (mg/L)	152.07 ± 19.41	127.20 (42.70–322.20)	289.70 ± 36.06 ^A,B^	240.20 (123.40–970.10)	180.2 ± 19.60	170.20 (52.01–377.80)
Mg (mg/L)	6.31 ± 0.59	5.84 (2.38–12.36)	7.67 ± 1.013	6.04 (2.21–20.95)	6.125 ± 0.78	5.25 (1.39–15.82)
Ba (μg/L)	1.13 ± 0.15	1.01 (0.29–2.14)	1.57 ± 0.12	1.33 (0.98–2.15)	0.83 ± 0.11	0.82 (0.10–0.86)
K (mg/L)	1025.27 ± 61.59	934.27 (684.60–1758.47)	967.0 ± 66.30	940.08 (382.70–1714.47)	986.0 ± 91.25	903.65 (423.90–2081.87)
Ca (mg/L)	30.05 ± 2.25	21.37 (13.09–49.56)	28.33 ± 2.74	27.96 (9.39–55.10)	26.58 ± 2.35	24.71 (7.09–52.46)
Mn (μg/L)	3.52 ± 0.58	2.69 (0.02–2.63)	7.29 ± 1.15 ^A,B^	5.59 (1.52–21.93)	4.00 ± 1.04	2.59 (0.76–23.82)
Fe (μg/L)	9.74 ± 1.47	9.12 (1.53–25.14)	26.66 ± 5.62 ^A,B^	15.57 (5.04–54.01)	12.56 ± 2.96	8.58 (1.22–28.92)
Cu (μg/L)	11.95 ± 1.86	8.89 (3.85–38.70)	31.58 ± 15.03	13.99 (3.78–73.76)	9.65 ± 1.66	7.23 (1.67–29.96)
Zn (μg/L)	46.01 ± 6.28	40.59 (3.78–95.57)	55.45 ± 11.12	46.13 (2.44–88.46)	29.42 ± 5.16 ^A^	23.00 (8.68–81.77)

Superscript A = *p* < 0.05 compared to healthy group. Superscript B = *p* < 0.05 compared to treated periodontitis group.

**Table 3 biomedicines-08-00354-t003:** Correlation between mineral elements in saliva of untreated periodontitis patients.

	Li	Na	Mg	K	Ca	Mn	Fe	Cu	Zn	Ba
Li	1.000	0.096	−0.041	0.267	0.239	−0.196	−0.164	−0.256	−0.198	0.229
Na		1.000	0.457 *	0.222	0.369	0.063	0.177	0.483 *	0.309	0.016
Mg			1.000	0.362	0.453 *	0.304	0.139	0.673 **	0.277	0.403
K				1.000	0.161	0.116	0.274	0.046	0.475 *	0.144
Ca					1.000	−0.044	−0.018	0.440 *	0.203	0.361
Mn						1.000	−0.363	0.527 **	0.229	−0.108
Fe							1.000	−0.083	0.150	0.048
Cu								1.000	0.352	0.342
Zn									1.000	0.159
Ba										1.000

* *p* < 0.05; ** *p* < 0.01.

**Table 4 biomedicines-08-00354-t004:** Correlation between mineral elements in saliva of treated periodontitis patients.

	Li	Na	Mg	K	Ca	Mn	Fe	Cu	Zn	Ba
Li	1.000	−0.308	−0.421	−0.127	0.137	−0.280	−0.451 *	−0.244	−0.329	0.145
Na		1.000	0.690 **	0.557 **	0.517 *	0.456 *	0.495 *	0.713 **	0.166	0.423
Mg			1.000	0.609 **	0.352	0.223	0.734 **	0.596 **	0.619 **	0.241
K				1.000	0.336	0.323	0.265	0.355	0.256	0.454
Ca					1.000	0.471 *	0.365	0.622 *	0.092	0.310
Mn						1.000	0.199	0.446 *	0.386	0.046
Fe							1.000	0.493 *	0.463 *	0.283
Cu								1.000	0.420	0.269
Zn									1.000	0.128
Ba										1.000

* *p* < 0.05; ** *p* < 0.01.

**Table 5 biomedicines-08-00354-t005:** Correlation between mineral elements in saliva of healthy controls.

	Li	Na	Mg	K	Ca	Mn	Fe	Cu	Zn	Ba
Li	1.000	0.344	0.081	0.107	0.259	0.069	−0.356	0.491 *	−0.274	0.148
Na		1.000	0.513 *	0.414	−0.077	0.117	0.239	0.573 **	0.323	−0.051
Mg			1.000	0.669 **	0.367	0.251	0.323	0.558 *	0.526 *	−0.93
K				1.000	0.062	0.337	0.099	0.550 *	0.349	0.008
Ca					1.000	0.272	−0.179	0.319	0.131	0.399
Mn						1.000	0.182	0.244	0.129	0.033
Fe							1.000	0.015	0.433	−0.223
Cu								1.000	0.242	−0.163
Zn									1.000	−0.291
Ba										1.000

* *p* < 0.05; ** *p* < 0.01.

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
