# Peer review of "ICP-Mass-Spectrometry Ionic Profile of Whole Saliva in Patients with Untreated and Treated Periodontitis"

_biomedicines, 2020, doi:10.3390/biomedicines8090354_

Round 1

Reviewer 1 Report

This paper describes comparison of mineral elements in saliva among untreated and treated periodontitis patients, and healthy volunteers. The authors concluded that Cu, Na, Fe, and Mn are significantly increased in saliva of severe periodontitis-affected patients. This is an interesting study but the results are predictable. The reviewer would like to mention a number of issues to be addressed, as noted below, for further study.

  1. There are no description on bacterial effects on the results. Please refer to Goodson (Periodontol 2000, 2003, doi: 10.1034/j.1600-0757.2003.03104.x), and discuss this point.
  2. In periodontitis patients, amount of gingival crevicular fluid are increasing (in above Goodson’s review), and therefore, Na in saliva may reflect this. Please include this in discussion section.
  3. As the authors stated, Fe in periodontitis patients may reflect occult blood hemoglobin in saliva. Please refer to Suzuki et al. (J Oral Sci, 2020, doi: 10.2334/josnusd.18-0226), and discuss this.
  4. A copper carrying protein and has ferroxidase activity, ceruloplasmin, has been increased in serum of periodontitis patients (Mohammed et al., Stomatological Dis Sci, 2018, doi: 10.20517/2573-0002.2017.15). Please describe this point in discussion section.
  5. Zinc-dependent MMPs are known to increase in saliva of periodontitis patients (for example, Maciejczyk et al., Adv Clin Exp Med, 2016, doi: 10.17219/acem/30428). Please discuss the reason for increasing Zn.
  6. The most important problem of this manuscript is validation experiments are lacking. Please construct the algorithm for diagnosis and validate this using new saliva samples. This is essential for practical use of this method in periodontal diagnosis.

Author Response

Dear Editor,

Thank you for the comments regarding our manuscript (biomedicines-903876) entitled “ICP-mass-spectrometry ionic profile of whole saliva in patients with untreated and treated periodontitis.” by F. Romano et al. The Reviewers comments spurred us to further improve the overall quality of our message.

Yours faithfully

Giovanni N Berta

Authors’ responses to Reviewer #1:

Reviewer #1:

Comment 1 to the Authors: There are no description on bacterial effects on the results. Please refer to Goodson (Periodontol 2000, 2003, doi: 10.1034/j.1600-0757.2003.03104.x), and discuss this point.

Authors’ response/action: We are grateful with the Reviewer to arise this point. We discussed this aspect between lines 379 and 381 in the text:

“It should be also considered that we could not discriminate between host cells and bacteria related products. Saliva accumulates elements of the metabolism of both bacterial and host origin from the GCF”.

Comment 2 to the Authors: In periodontitis patients, amount of gingival crevicular fluid are increasing (in above Goodson’s review), and therefore, Na in saliva may reflect this. Please include this in discussion section.

Authors’ response/action: Thanks for this remark. We deepened this aspect between lines 259 and 263 in the discussion:

“It has been also reported that Na concentration tends to increase proportionally according to the severity of the inflammatory status [38]. The volume of GCF increases as inflammation becomes more severe and periodontal damage worsens [39], thus higher salivary Na concentration may reflect the destructive inflammatory reaction within periodontal environment”.

Comment 3 to the Authors: As the authors stated, Fe in periodontitis patients may reflect occult blood hemoglobin in saliva. Please refer to Suzuki et al. (J Oral Sci, 2020, doi: 10.2334/josnusd.18-0226), and discuss this.

Authors’ response/action: Thanks for raising this issue. We clarified this point between the lines 289 and 294 in the discussion:

“The concentration of hemoglobin in saliva or oral rinse using water has demonstrated positive association with degree of gingival inflammation, periodontal parameters, and bone loss [46]. For these reasons the salivary occult test has been proposed for periodontal diagnosis, but has not gained widespread use due to the false positive results in healthy subjects and the inability to selectively detect periodontitis patients [47]”.

Comment 4 to the Authors: A copper carrying protein and has ferroxidase activity, ceruloplasmin, has been increased in serum of periodontitis patients (Mohammed et al., Stomatological Dis Sci, 2018, doi: 10.20517/2573-0002.2017.15). Please describe this point in discussion section.

Authors’ response/action: This is a very interesting question. In the revised version, we discussed it. In particular, we added a new phrase in the discussion section between lines 306 and 309: “Interestingly, serum levels of ceruloplasmin, that is an acute phase reactant with ferroxidase activity and involved in Cu transport, appear significantly elevated in periodontitis when compared with those from healthy controls [51]. This finding may suggest a defensive response to the oxidative stress induced by periodontal inflammation”.

Comment 5 to the Authors: Zinc-dependent MMPs are known to increase in saliva of periodontitis patients (for example, Maciejczyk et al., Adv Clin Exp Med, 2016, doi: 10.17219/acem/30428). Please discuss the reason for increasing Zn.

Authors’ response/action: Thanks to the Reviewer for raising this issue. We discussed it between lines 332 and 335 in the new version:

“Furthermore, periodontal treatment reduces the salivary levels of zinc-dependent MMPs, tissue-destructive enzymes that are responsible for the extracellular matrix degradation in periodontitis. Controlling these inflammatory mechanisms contributes to inhibiting the progressive tissue breakdown and collagen depletion in the periodontal tissues [56]”.

Comment 6 to the Authors: The most important problem of this manuscript is validation experiments are lacking. Please construct the algorithm for diagnosis and validate this using new saliva samples. This is essential for practical use of this method in periodontal diagnosis.

Authors’ response/action: We are in agreement with what has been stated. Unfortunately, in a maximum of 10 working days to carry out the revision of the manuscript, it is not possible to perform new experiments on saliva samples. However, our work is preliminary and to be implemented for the clinical translational aspect. Nevertheless, this raised aspect was included in the discussion as a limitation of our study in lines 383and 384:

“A larger sample size would also enable the development of a clinical algorithm. We intend to extend the research to achieve this goal within the next future”.

Reviewer 2 Report

The manuscript contains some novel ideas and concepts.

The abstract is relevant to the content and includes proper key words. The study group was  selected carefully.

The chapter "Material and methods" is described in detail. The conclusions are supported by the data. Results are corresponding to material and methods and are clearly stated.

Figures illustrate research well, legends are correct. References are appropriate to the text but they should be more up-to-date. Citation in the text is correct.

Author Response

Thank you for the revision of our manuscript

Giovann N Berta

Round 2

Reviewer 1 Report

The manuscript has fairly been improved.